# Can Socioeconomic, Health, and Safety Data Explain the Spread of COVID-19 Outbreak on Brazilian Federative Units?

**DOI:** 10.3390/ijerph17238921

**Published:** 2020-11-30

**Authors:** Diego Galvan, Luciane Effting, Hágata Cremasco, Carlos Adam Conte-Junior

**Affiliations:** 1COVID-19 Research Group, Center for Food Analysis (NAL), Technological Development Support Laboratory (LADETEC), Cidade Universitária, Rio de Janeiro RJ 21941-598, Brazil; conte@iq.ufrj.br; 2Laboratory of Advanced Analysis in Biochemistry and Molecular Biology (LAABBM), Department of Biochemistry, Federal University of Rio de Janeiro (UFRJ), Cidade Universitária, Rio de Janeiro RJ 21941-909, Brazil; 3Nanotechnology Network, Carlos Chagas Filho Research Support Foundation of the State of Rio de Janeiro (FAPERJ), Rio de Janeiro RJ 20020-000, Brazil; 4Chemistry Department, State University of Londrina (UEL), Londrina PR 86057-970, Brazil; luciane.effting@uel.br (L.E.); hagata@uel.br (H.C.)

**Keywords:** coronavirus disease, artificial neural networks, SARS-CoV-2, ventilator, index development index, developing country

## Abstract

Infinite factors can influence the spread of COVID-19. Evaluating factors related to the spread of the disease is essential to point out measures that take effect. In this study, the influence of 14 variables was assessed together by Artificial Neural Networks (ANN) of the type Self-Organizing Maps (SOM), to verify the relationship between numbers of cases and deaths from COVID-19 in Brazilian states for 110 days. The SOM analysis showed that the variables that presented a more significant relationship with the numbers of cases and deaths by COVID-19 were influenza vaccine applied, Intensive Care Unit (ICU), ventilators, physicians, nurses, and the Human Development Index (HDI). In general, Brazilian states with the highest rates of influenza vaccine applied, ICU beds, ventilators, physicians, and nurses, per 100,000 inhabitants, had the lowest number of cases and deaths from COVID-19, while the states with the lowest rates were most affected by the disease. According to the SOM analysis, other variables such as Personal Protective Equipment (PPE), tests, drugs, and Federal funds, did not have as significant effect as expected.

## 1. Introduction

In December 2019, cases of acute pneumonia in the city of Wuhan, China, called attention due to the speed of contagion [1]. The virus that caused this pneumonia in affected individuals is the SARS-CoV-2, the causative agent of the novel coronavirus (COVID-19). The number of cases quickly evolved into a pandemic, decreed by the World Health Organization (WHO) on 11 March 2020. Currently, a few months after the first cases, this disease has been plaguing the world [2].

At the moment, the pandemic epicenter is in Latin America, with Brazil, Argentina, Colombia, Peru, Mexico, and Chile being the most representative countries, concentrating more than 10 million cases. Brazil already has more than 5 million cases and 150,000 deaths, ranking third in the world in the number of cases and deaths by COVID-19, behind only to the United States and India [3].

Intense efforts by the scientific community are focused on finding drugs to treat [4] and develop a vaccine for the disease [5]. Other research tries to understand how the disease has spread around the world. Some factors can influence the spread of the virus, including the disregard of preventive measures adopted by the government by population [6], climatic [7,8,9], comorbidities [10,11], hospital structure, prepared professionals, personal protective equipment, and financial resources [4].

In order to monitor the pandemic, it is necessary to collect information on the most significant possible number of factors that can influence the disease’s dissemination behavior and, with this, establish a relationship that allows to affirm which measures delay the spread of COVID-19. Statistical tools have been used to study the behavior of epidemics for years; this area treats, analyzes, and obtains information from large datasets that cannot be analyzed by traditional systems [12].

Artificial Neural Networks (ANNs) are tools that have been gaining scientific relevance to perform pattern recognition, classification, or prediction tasks. ANN is a non-linear computational model attempting to simulate human brain structure and decision-making [13]. Its architecture is inspired by biological neural networks and consists of simple processing units that store empirical knowledge through a learning process [14].

Among the most common types of neural networks are the Feed-Forward Neural Network (FFNN), the Convolutional Neural Network (CNN), and the Recursive Neural Network (RNN). FFNNs are the most common type of ANN in practical applications, consisting of one or more hidden layers of perceptrons (neurons) that require supervised training. The input data of the desired sample datasets and the output results are sent to the network several times until the error in the output is minimized [13].

Traditionally, several types of neural networks exist with various techniques such as Autoregression (AR) [15,16], Moving Average (MA) [17,18], Exponential Smoothing (ES) [19], Hybrid Methods (HM) [20,21,22], and Autoregressive Integrated Moving Average (ARIMA) [23]. They have been used to predict the dependent variable in a time series [15,16,17,18,19,20,21,22,23]. Among these techniques, unsupervised neural networks of the type Self-Organizing Maps model has mostly outperformed others in precision and accuracy [14].

Self-Organizing Maps (SOM) or Kohonen Map is a method for the analysis of multivariate data used for pattern recognition and classification, which may be taken as a non-linear generalization of principal component analysis [24]. The SOM algorithm consists of input nodes and a grid of computational connected nodes (neurons), which compete among themselves for activation as the one that most closely resembles the input vector. If the input data exhibit some similarity across the input classes, the neurons will organize themselves, showing similarity patterns in a grid [13,25].

SOM has already been used successfully for data mining in several areas of knowledge [24,26,27,28,29,30], like food science [26,28,29], fuels [27], monitoring chemical reactions [24,30], and other applications [25,31,32]. Recently, SOM was used to verify the spatial relationship of the COVID-19 spread in countries and states in Mexico [31]. Most published papers have the main theme of the modelling or prediction of the spread of COVID-19 [2,9,33,34,35,36]. In our literature searches, we did not evidence reports that simultaneously consider the historical data on the number of cases/deaths from COVID-19 with possible external factors that affect the spread of the virus through a spatial analysis by pattern recognition.

In a previous study, we demonstrated that the spread of COVID-19 varies according to Brazilian regions, states, and cities. However, it was not possible to allege the reason for this behavior. In this study, 14 possible factors that can affect the spread of COVID-19 were analyzed jointly by SOM. This analysis will verify which variables may be essential and will point out possible variables that had the most significant effect on each Brazilian state.

## 2. Theoretical Foundations

The Kohonen algorithm was proposed by Teuvo Kohonen in 1982 [14]. In SOM, there is no given output target, the objective of the algorithm is to find a set of neurons to represent the cluster, but with topological restrictions [31]. The learning process begins with the random initialization of the synaptic weights vector of each neuron. In sequence, three key steps are developed for the formation of the feature map: Competition, cooperation, and synaptic adaptation [24].

The following is a brief description of the Kohonen algorithm, as described previously by Haykin [14]. The function is chosen to represent the topological neighborhood in the following equation.
$$h_{j,i}=\exp\left(-d_{j,i}^{2}/2\sigma^{2}\right)$$
where σ is the effective radius of the topological neighborhood, and *d_j,i_* is the lateral distance between the winning neuron *i* and the excited neuron *j*, defined through Euclidean distance. Over the training epochs, there is a reduction in the size of the neighborhood due to an exponential decay, described by the equation.
$$\sigma \left(n\right)=\;\sigma_{0}\exp\left(-n/\tau_{1}\right)\;\;\;n=0,1,2,\ldots$$
where σ_0_ is the effective radius in the initialization of the algorithm, τ_1_ is the time constant, with τ_1_ = 1000/log σ_0_ being recommended, and *n* is the number of training epochs.

During the adaptive process, the synaptic weight vector (**w***_j_*) of the *j* neuron in the grid must be modified concerning the input vector **x**. The modification process is a modification of the Hebb learning postulate, described by the equation.
$$w_{j}\left(n+1\right)=\;w_{j}\left(n\right)+\left(\eta \right)h_{j,i\left(x\right)}\left(x-w_{j\left(n\right)}\right)$$
where *η*(*n*) is the learning rate, which is variable and decreases during the training epochs, *n*. The learning rate decrease may be modelled by an exponential decay, as described in the equation. In this equation, *η*_0_ is the initial learning rate, and τ_2_ is another time constant; the recommended values are, respectively, 0.1 and 1000:$$\eta \left(n\right)=\;n_{0}\exp\left(-n/\tau_{2}\right)$$

## 3. Methodology

### 3.1. Dataset

The dataset used in the study by neural networks was obtained from websites of Brazilian government agencies or institutions, available for download. Before the SOM analysis, the variables that would constitute the study were selected. All variables related to socioeconomic, health, and safety factors that could explain the dissemination of COVID-19 in Brazil were included in our research. They were made available and updated daily by institutional sites of the Brazilian federal government.

Initially, data for all variables were converted and expressed as rates for each 100,000 inhabitants. For the conversion, we used data from the Brazilian population estimated in 2019 by the Instituto Brasileiro de Geografia e Estatística – IBGE (Brazilian Institute of Geography and Statistics); the public agency responsible for conducting censuses and organizing information related to the country’s geosciences and social, demographic, and economic statistics [37]; available in https://www.ibge.gov.br/.

The numbers of cases and accumulated deaths by COVID-19 were analyzed for 16 epidemiological weeks. According to the international convention, each week starts on Sunday and ends on Saturday. The information represents the numbers recorded from 26 February to 13 June, 2020 (110 days), a period that corresponds from the 9th to the 24th epidemiological week of the year, first COVID-19 case record in Brazil. The data were obtained from the website of the Ministério da Saúde—MS (Ministry of Health) of Brazil, the government sector responsible for the administration and maintenance of public health in the country, updated daily with information about the COVID-19 pandemic in the country [38]; available in https://covid.saude.gov.br/.

The Federal Government of Brazil transfers financial resources, supplies, and equipment for structuring health services in the country to combat or treat patients affected by COVID-19. The MS frequently updates a panel of inputs considered essential, distributed to each Brazilian state: Influenza vaccination—H1N1 and H3N2 (distributed and applied), drugs (chloroquine and oseltamivir), COVID-19 tests (rapid and reverse-transcriptase polymerase chain reaction—RT-PCR), hand sanitizer per liter, and Personal Protective Equipment—PPE (surgical masks, N95 mask, gowns, gloves, glasses, face shield, caps, and sneakers) [39]. These data were obtained on June 13, 2020, at https://covid-insumos.saude.gov.br/paineis/insumos/painel.php.

The destination of Federal funds transferred to each federative unit (Brazilian states) is monitored and presented on the Federal Government website [40], available in https://www.tesourotransparente.gov.br/visualizacao/painel-de-monitoramentos-dos-gastos-com-covid-19. The data used in this study represent the sum of all transfers made until 13 June 2020. For better visualization, the values were converted from the real (R$) to the US dollar (US$) by the quotation of R$ 5.05 equals to US$ 1.00, referring to 12 June 2020, provided by the Banco Central do Brasil—BCB (Central Bank of Brazil) [41], available in https://www.bcb.gov.br/.

Health data from public and private institutions, such as the number of physicians, nurses, ventilators, and Intensive Care Unit (ICU) beds for COVID-19 in each state, were obtained from the IBGE [42]; available in https://mapasinterativos.ibge.gov.br/covid/saude/.

Finally, the data referring to the Human Development Index (HDI) of each state for the year 2015 (last census registered in the country) were obtained from the Atlas do Desenvolvimento Humano do Brasil—ADHB (Human Development Atlas of Brazil), a public institution that provides information about human development in Brazil [43]; available in http://www.atlasbrasil.org.br/2013/pt/download/base/.

### 3.2. Data Analysis by Artificial Neural Network

The Kohonen Map routine developed was carried out in Matlab software (MathWorks, Natick, MA, USA) according to the algorithm previously described in Haykin [14] and Cremasco et al. [26]. The datasets were evaluated using three different approaches. In the first step, two SOM analyses were conducted to verify the distribution of the numbers of cases and deaths from COVID-19 in Brazilian regions and states. In the second step, the distribution of 14 variables for the Brazilian states was verified: ICU beds, ventilators, physicians, nurses, PPE, hand sanitizer, rapid test, PCR test, vaccines distributed, vaccines applied, chloroquine tablets, oseltamivir capsules, HDI, and federal funds distribution.

The SOM setup was a hexagonal topology of 8 × 8 and 4 × 4 for 27 Brazilian states and five regions, respectively, for the number of cases and deaths from COVID-19 (**first step**); 8 × 8 for 27 Brazilian states with 14 variables (**second step**) with 7000 training epochs to ensure convergence of the average quantization error. The initial neighborhood relationship was 3.5, decreasing to 0.07, with an initial learning rate of 0.1, decaying exponentially with the training epochs to 9.11 × 10^−5^. It was used a computer Intel^®^ Core™ i7–4790 CPU^©^ 3.60 GHz, 32 GB RAM, and 250 GB HDD.

In order to obtain a better representation, the values collected after the SOM analysis were transposed through the color scale to the Brazilian cartographic map for each variable. This procedure was adopted to facilitate the interpretation of unfamiliar people with the Kohonen map. The original SOM output weight maps for the 14 variables generated are displayed in the Appendix A.

### 3.3. Spearman’s Correlation Test

The Spearman’s rank correlation coefficient is adopted to determine the correlation between variables. It analyzes how well the association between two variables can be defined using a monotonic function. The correlation matrix was performed using the package “corrplot” in the Software R Core Team (Vienna, Austria) [44].

## 4. Tests and Results

Figure 1 shows the number of cases and deaths accumulated by Brazilian states and regions per 100,000 inhabitants. All states belonging to the South (S) and Central-West (CW) regions of the country had the lowest rates of cases and deaths by COVID-19 recorded, an average of 180 cases and 4 deaths per 100,000 inhabitants. Most of the Brazilian states with the highest rates of cases and deaths belong to the North (N) region, an average of 954 cases and 43 deaths per 100,000 inhabitants, mainly represented by Acre (AC), Amapá (AP), Amazonas (AM), Pará (PA), and Roraima (RR).

Some states in the Brazilian Northeast (NE) and Southeast (SE) also had high rates of cases and deaths when compared to the other states in the South (S) and Central-West (CW) regions. The Northeast (NE) is the second region with the highest rate of cases and deaths, an average of 527 cases and 24 deaths per 100,000 inhabitants, the states in this region that were most aggravated by the disease were Ceará (CE), Maranhão (MA), and Pernambuco (PE). The Southeast (SE) is the third region with the highest rate of cases and deaths, with an average of 338 cases and 23 deaths by COVID-19, with Espírito Santo (ES), Rio de Janeiro (RJ), and São Paulo (SP) being the most representative states in the region.

The neural network demonstrated that the spread of COVID-19 in Brazil has heterogeneous behavior; see Figure 1. It is essential to understand this behavior. The North of the country was more affected by COVID-19 than the South. The Northeast and Southeast differ in case rates but have similar death rates. In this sense, possible factors were evaluated that may explain the reason for this behavior and point out which measures are more relevant in the fight against COVID-19 in each Brazilian state.

After the training phase of the SOM network, we generated the topological map of the 14 variables evaluated together, which represents the distribution of each federative unit in Brazil, according to the winning neuron; see Figure 2. In the topological map, each federative unit is associated with a respective winning neuron, that is, the one that best represents in the analysis. The SOM network classifies the input data as clusters that can be formed by one or more neurons. The definition of clusters is characterized by the presence of empty neurons between the groups. Nearby clusters share some similarity, that is, the greater the Euclidean distance, the greater the difference in behavior.

We identified the formation of some clusters by evaluating the topological map. Among some of the clusters formed, we highlight the one formed in the lower-left corner of the map in Figure 2, represented by the states of the Southern region of the country, which contain the federative units: Paraná (PR), Rio Grande do Sul (RS), and Santa Catarina (SC). This cluster contains the federal units and the region of the country with the lowest rates of deaths and cases by COVID-19, as shown in Figure 1. Another cluster formed that we can highlight is composed of some states in the North region of the country, represented by the states of Acre (AC), Amapá (AP), and Roraima (RR), presented in the upper-right corner of the map. This cluster represents the region and some of the federative units with the highest rates of deaths and cases due to COVID-19 in the country. This same analysis can be done for the other formed clusters.

To verify the applicability of SOM analysis, we compared the results obtained with another unsupervised method. Thus, we evaluated the ability to group the data obtained using the Hierarchical Cluster Analysis (HCA) method. The results obtained with the HCA were very similar to those obtained with the SOM algorithm. The dataset presents different patterns according to the region in which the Brazilian states are located. In other words, most states in the North and Northeast regions formed a cluster, while most federative units in the Central-West, South, and Southeast formed another cluster. The graphic output generated by the HCA method is shown in Appendix A.

In general, the topological map of the SOM network made it possible to state that the socioeconomic, health and safety data demonstrate that the spread of COVID-19 in the country varies according to the Brazilian federative units. However, only the topological map does not allow to state which of the 14 variables evaluated may be the main responsible for explaining the spread of COVID-19 in the country. Thus, we used weight maps in the following discussions. In order to obtain a better representation, the values collected from the weight maps were transposed through the color scale to the Brazilian cartographic map for each variable. However, when doing the transposition procedure, we lose the neighborhood relationship. In Appendix A, the original outputs of the weight maps for each variable in the SOM network are shown.

The 14 variables were evaluated jointly by ANN. However, for better results visualization and discussion, the weight maps were divided into four blocks and grouped according to similarities: (i) Available hospital infrastructure; (ii) inputs and tests available; (iii) drugs available; and (iv) financial resources,

The first set of variables is shown in Figure 3, composed of the rates of ICU beds, ventilators, physicians, and nurses per 100,000 inhabitants. In general, it is observed that the states that belong to the South, Southeast, and Central-West regions of Brazil had higher rates than the North and Northeast. There is evidence that these variables influence the rates of cases and deaths in each Brazilian state, and it can be said that the states with the lowest rates of ICU, respirators, physicians, and nurses, have the highest rates of cases and deaths from COVID-19.

It is also evident that these were not the only relevant factors. The state of Rio de Janeiro (RJ), for example, has high rates of ICU beds, respirators, physicians, and nurses and despite this, it has a higher death rate than other states in the North region, such as Acre (AC), Amapá (AP), and Roraima (RR).

The second set of variables presented in Figure 4 represents the inputs destined to each state by the Federal Government to combat and control the spread of COVID-19, composed by PPE, hand sanitizer, rapid test, and PCR test per 100,000 inhabitants. In general, the PPE and hand sanitizer rates distributed for each state did not have a direct relationship with the rates of cases or deaths by COVID-19 according to the SOM analysis.

The states of Amapá (AP) and Rio de Janeiro (RJ), for example, had the highest rates of PPE distributed, and Amazonas (AM) had the highest rates of liters of hand sanitizer distributed per 100,000 inhabitants; however, these three states present higher rates of cases and deaths when compared to other Brazilian states.

It is important to note that the data refer to the distributed quantity of inputs, and it is not possible to estimate which portion was used by hospitals or the population, or even say if they were used correctly. The cultural factor also interferes in these variables, mainly concerning the education and awareness of the local population about the use of these inputs [45].

The disease tests in the country have been carried out homogeneously, as shown in Figure 4. This homogeneity is considered an important aspect and demonstrates that differences do not influence the data collected regarding the number of cases in the form of confirmation of the disease, which could question the reliability of the data. The states that had the highest test rates available were the Distrito Federal (DF), Rio Grande do Sul (RS), and Roraima (RR), which belong to the Central-West, South, and North regions, respectively, i.e., the most and least affected regions by COVID-19 in the country.

Rapid tests are cheaper, less reliable, and can result in false positives or negatives, while the PCR test is reliable but expensive and time-consuming [46]. It is necessary to adopt a balance between rapid tests and PCR; for example, if PCR test indices are higher, it will not be possible to follow the spread in a short period, while rapid tests, when misused, can generate mistaken information.

Previous measures as vaccines and drugs against diseases with similar symptoms, such as influenza H1N1 and H3N2, can facilitate the diagnosis of COVID-19, since they are relevant information during the patient’s anamnesis. These procedures allow faster and more efficient identification of the disease, which makes the adopted treatments more assertive and precocious [47]. Another measure that focuses on significant discussions in the scientific community is the use of drugs against the coronavirus [48]. Among several drugs, the Brazilian Federal Government has invested and passed on chloroquine tablets to the states.

The third set of variables in Figure 5 consists of influenza vaccines distributed and applied, and drugs distributed by states per 100,000 inhabitants. The states of the North region, such as Amazonas (AM), Roraima (RR), Pará (PA), and Acre (AC), had a more significant disparity than the other states in the rates of vaccines applied and distributed. At first, this behavior could be related to health factors, shown in Figure 3. However, this disparity would also be observed in the Northeast region. Other intrinsic factors may include geographic and logistical factors that make distribution and access to the population a challenge to receive the vaccine.

As far as we know, there is no drug with proven efficiency against SARS-CoV-2; some drugs have been evaluated to combat or alleviate the disease’s symptoms [48]. Recent studies demonstrate that dexamethasone welcomes the preliminary treatment of critically ill patients with COVID-19 [49]. Recently, the Food and Drug Administration (FDA) revoked emergency use authorization for chloroquine and hydroxychloroquine in patients with COVID-19 [50].

Figure 5 shows the chloroquine rates distributed by the Federal Government per 100,000 inhabitants. Chloroquine tablets were more widely distributed in the states of the North region, which has higher rates of deaths and cases of COVID-19 in Brazil. The possibility of greater distribution of chloroquine in these states may have occurred as an attempt to treat the disease symptoms when there were no restrictions on its use.

The oseltamivir rates distributed by the Federal Government per 100,000 inhabitants is shown in Figure 5. This drug is commonly used to treat symptoms caused by the influenza virus and is administered when there is still no confirmed diagnosis of COVID-19 [51]. Oseltamivir was more widely distributed in the Southern and Northern states of Brazil; however, it is not possible to establish a direct relationship with the rates of cases and deaths by COVID-19 using SOM analysis.

The last set of variables evaluated comprises the HDI and the distribution of federal funds destined to each state. According to Figure 6, the destination of Federal funds was more significant for some states in the North, the region which has the highest rates of cases and deaths due to COVID-19. More resources destined for this region can be guided by the scenario of cases and deaths registered.

The highest HDI values are in the Central-West, South, and Southeast regions, with the Federal District (DF), Santa Catarina (SC), and São Paulo (SP) being the highest values in the country. It is also worth mentioning that the most affected states by COVID-19 have some of the lowest HDIs in the country, showing problems that some of these states have in fundamental areas of the population service.

Figure 7 shows the Spearman’s correlation test adopted to determine the correlation among variables. The test demonstrated that the influenza vaccine applied, ICU, ventilators, physicians, nurses, and HDI were the most correlated significant variables, all positive correlations, while chloroquine had a negative correlation with all these variables. The HDI is directly related to essential aspects of the population, such as family income, education level, health, among other factors; therefore, the correlation with the HDI was already expected, since it takes into account important aspects of health for its determination. Other variables evaluated showed minor correlations, while PPE did not correlate with any other variable.

## 5. Discussion

Considering the importance of understanding the spread of the virus, many studies have been presented in recent months related to different aspects of the pandemic, with the application of computational intelligence tools to model and predict the spread of the disease [34].

In our study, we gave the data a differentiated approach from the reported works. It is not common to find simultaneous studies with historical data on the number of COVID-19 cases, related to external factors, that affect the spread of the virus through spatial analysis [31]. For this, we used unsupervised pattern recognition to analyze some variables, among many others, that may be related to the disease spread.

Brazil is one of the most affected countries by the pandemic and by the tremendous socioeconomic, territorial, climatic differences, among the country’s federative units. We understand that it may be interesting to try to establish spread patterns of the virus in the country. Thus, once the relationship of these factors with the spread and lethality of the disease has been identified, we hope that our study can assist in the analysis of the data that have been generated by institutions and, consequently, assist in directing decision-making on practical combat actions.

It is important to note that our analysis does not aim to point out or compare which were the best actions adopted to contain the spread of COVID-19, so we approached each variable according to the significance indicated by the SOM analysis.

Based on SOM’s clustering skills, we were able to spatially group similar federative units in terms of the number of cases and deaths by COVID-19 with data on the distribution of financial resources, equipment, health professionals and HDI, represented by a color scale. Thus, SOM’s ability to cluster, made it possible to cluster together with the federal units that are behaving similarly and, therefore, can benefit from similar strategies to deal with the virus spread.

The SOM analysis allowed us to raise some hypotheses about the spread of the virus in Brazil. In general, the analysis indicated that the spread of the disease has a direct relationship with quite heterogeneous socioeconomic, health, and safety variations in the national territory; see topological map in Figure 2.

In Figure 2, we can see that the spread of the disease in Brazilian states has been differentiated and regionalized. Interestingly, the network separated the most and least regions affected by COVID-19. We found that there was a separation pattern, in which all the Southern federative units are located at the bottom left of the map (PR, RS, and SC), while the Southeast states (SP, RJ, MG, and ES) and the Central-West (MS, MT, DF, and GO) are in the upper left corner of the map. In the North and Northeast regions, the federative units were classified on the right side of the map.

For better visualization, we invite readers to check the original output weight maps of the variables in the Appendix A. Weight maps represent the overlap of the topological map (Figure 2) and allow us to evaluate the behavior of each variable for the segmentation of the federative units. Analyzing the weight maps for the 14 variables extracted from the SOM network, it was possible to show which were the main responsible for differentiating the states, regarding the number of cases and deaths by COVID-19 in Brazil by socioeconomic, health, and safety data.

According to the 14 weight maps, the HDI was one of the essential variables for the distribution obtained (Figure 6 and Appendix A). The analysis indicated that the federal units with the lowest HDI were considerably more affected by the pandemic and that they had greater difficulty in combating the spread of the virus. Meanwhile, the best-prepared Brazilian federations with higher HDI values were more capable of deal with the pandemic.

Possibly, the HDI was the main measure responsible for the separation due to its high correlation with the other variables. It is noteworthy that this index relates to health (life expectancy), education (adult literacy index and levels of education), and income (GDP—gross domestic product—per capita). Thus, it is possible to observe the correlation of the HDI in Figure 7, with other variables, such as ICU (Appendix A), ventilators (Appendix A), doctors (Appendix A), and nurses (Appendix A).

Another important variable that allowed us to assess the behavior of the spread of COVID-19 in Brazil was the influenza vaccine rates of doses distributed and applied, highly correlated according to the Spearman correlation test (Figure 7). According to Figure 6, Appendix A, we show that the federative units with the highest rates of vaccines applied had lower rates of cases and deaths from COVID-19, as pointed out by the SOM analysis. Although this procedure is not effective against SARS-CoV-2, this measure may have facilitated clinical diagnosis and made the treatment of affected patients by COVID-19 faster and more accurate, and may have reflected in the number of cases and deaths in these regions and federative units.

Other measures adopted in the country, such as the use of drugs as chloroquine and oseltamivir, did not allow to evaluate the spread of COVID-19, as shown in Figure 5 and the weight maps of the variables in Appendix A. The SOM analysis indicated that the federative units in the North and Northeast regions had higher rates of chloroquine tablets distribution but more cases and deaths from the disease were registered. On the other hand, oseltamivir capsules had higher distribution rates in the Southern and Northern states, representing the regions more and less affected by the disease, which means that this variable, the spread of COVID-19 was similar.

We evidence that PPE (Appendix A) and hand sanitizer (Appendix A) rates distributed for each state did not have a direct relationship with the rates of cases or deaths by COVID-19 when considering the SOM analysis (Figure 4). It is known that these measures directly influence the control of the COVID-19 dissemination [45]; in this sense, we show that the SOM analysis allowed us to verify that although these items have been distributed to federative units, the population has not followed the measures adopted or made use of these items properly. Figure 7 reinforces this hypothesis since the PPE did not correlate with any other variable. Thereby, government agencies must further encourage the use of available items.

Among the other less expressive variables evaluated, we show that the rapid test (Appendix A), PCR test (Appendix A), and distribution of Federal funds (Appendix A) did not present an obvious behavior that would allow explaining the spread of COVID-19 using the numbers of cases and deaths in the country, according to the SOM analysis indicated.

## 6. Conclusions

Unsupervised Artificial Neural Networks of the Self-Organizing Map type have demonstrated that the spread of the coronavirus has heterogeneous behavior and varies among Brazilian regions and states. According to the analysis, among the 14 variables evaluated, the factors responsible for the highest relationship with the numbers of cases and deaths by COVID-19 in Brazilian states were: Rates of influenza vaccine applied, ICU beds, ventilators, physicians, nurses, and HDI, positively correlated according to the Spearman’s correlation test. In general, the lowest rates of cases and deaths by COVID-19 were recorded in the Brazilian states with the highest rates of influenza vaccine applied, ICU beds, respirators, physicians, and nurses, per 100,000 inhabitants, which consequently has some of the highest HDI in the country.

As a future work, we intend to integrate the SOM algorithm to analyze the spatial and temporal aspects of the COVID-19 spread in a unified way to obtain a complete view and solution to the problem. In addition, to analyze the possibility of applying SOM in other diseases that have affected Brazil, caused by dengue virus, Chikungunya virus, Zika virus, and yellow fever virus.

## Figures and Tables

**Figure 1 ijerph-17-08921-f001:**
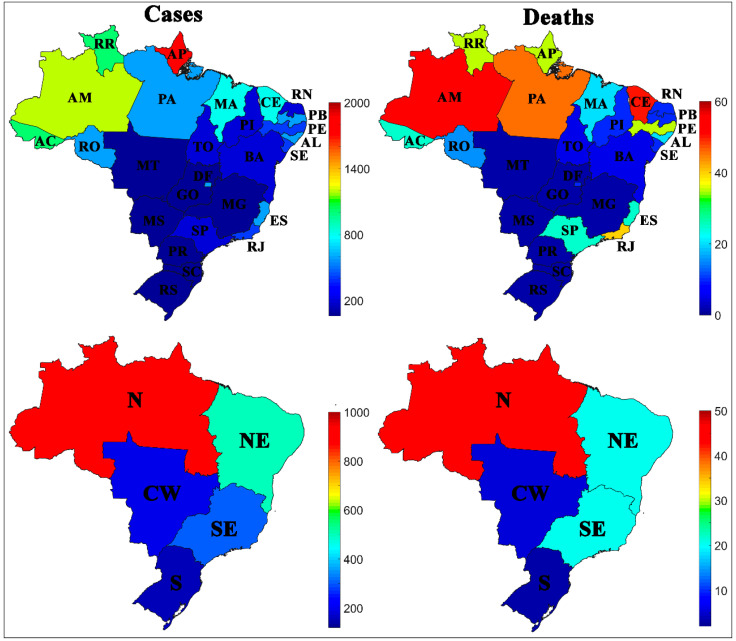
Geographical distribution of cases and deaths by COVID-19 in Brazilian states (above) and regions (below) per 100,000 inhabitants.

**Figure 2 ijerph-17-08921-f002:**
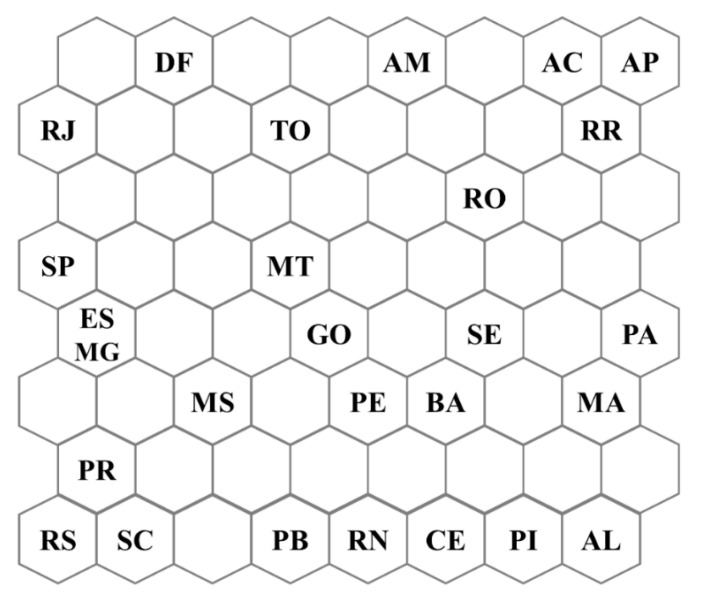
Distribution of Brazilian federative units according to the winning neuron. Where: Acre—AC; Alagoas—AL; Amapá—AP; Amazonas—AM; Bahia—BA; Ceará—CE; Distrito Federal—DF; Espírito Santo—ES; Goiás—GO; Maranhão—MA; Mato Grosso—MT; Mato Grosso do Sul—MS; Minas Gerais—MG; Pará—PA; Paraíba—PB; Paraná—PR; Pernambuco—PE; Piauí—PI; Roraima—RR; Rondônia—RO; Rio de Janeiro—RJ; Rio Grande do Norte—RN; Rio Grande do Sul—RS; Santa Catarina—SC; São Paulo—SP; Sergipe—SE; Tocantins—TO.

**Figure 3 ijerph-17-08921-f003:**
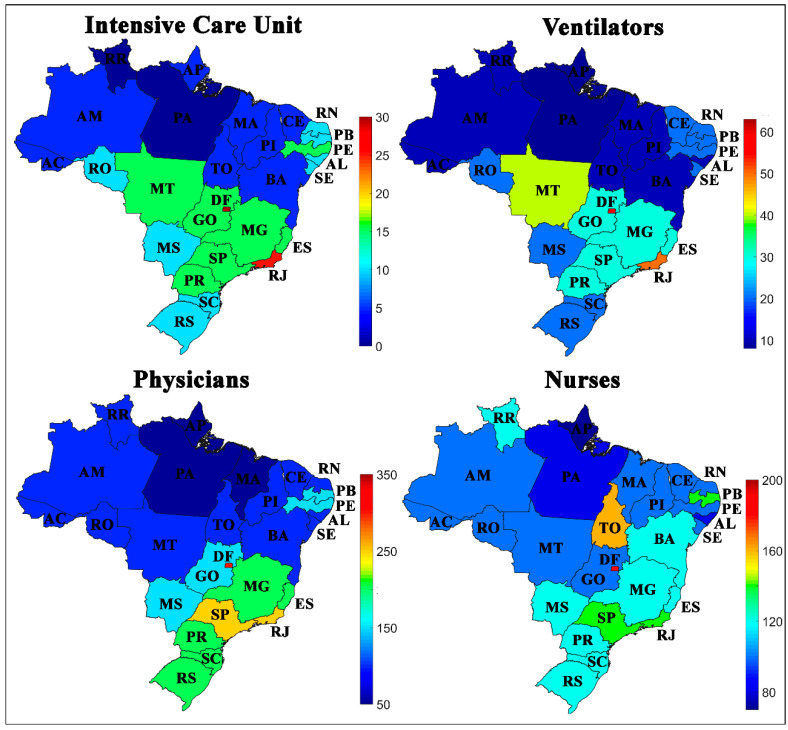
Geographical distribution of Intensive Care Unit (ICU) bed, ventilators, physicians and nurses in Brazilian states per 100,000 inhabitants.

**Figure 4 ijerph-17-08921-f004:**
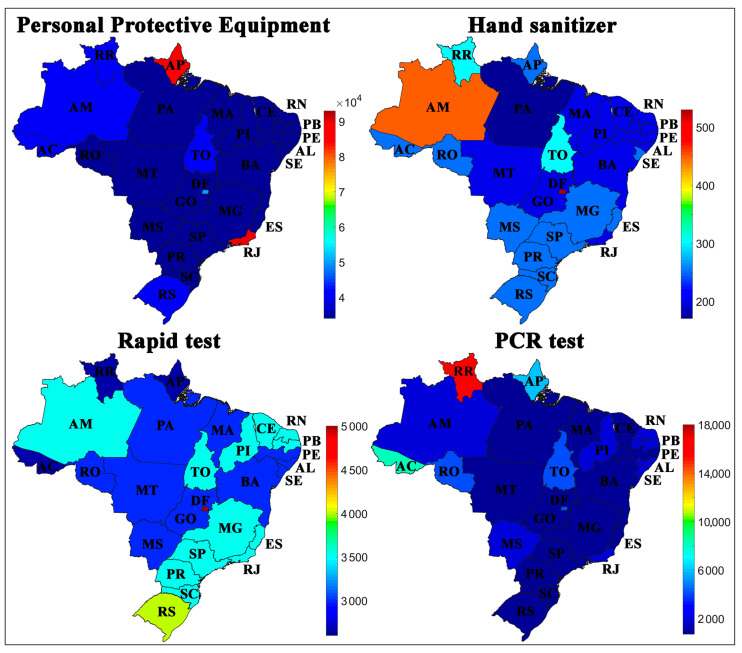
Geographical distribution of Personal Protective Equipment (PPE), hand sanitizer, rapid tests, and PCR tests available to Brazilian states per 100,000 inhabitants.

**Figure 5 ijerph-17-08921-f005:**
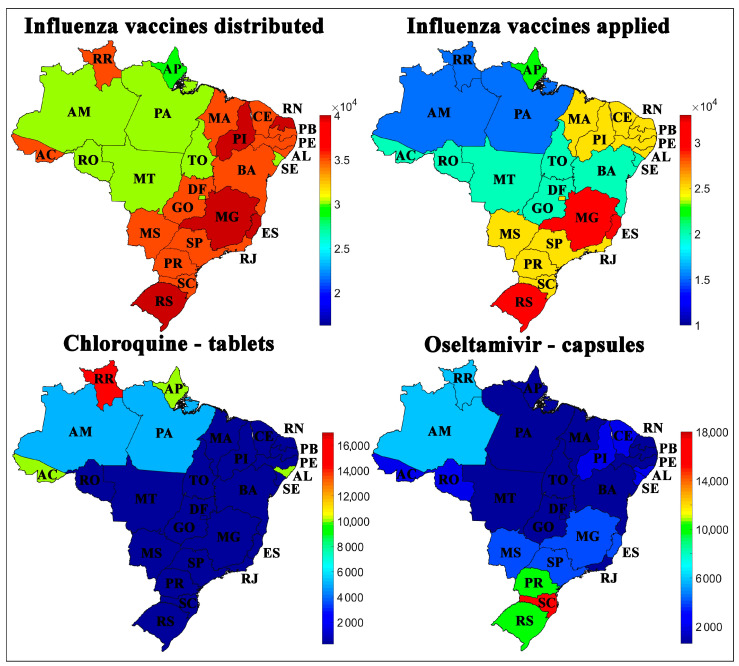
Geographical distribution of influenza vaccines distributed, influenza vaccines applied, chloroquine tablets, and oseltamivir capsules available in Brazilian states per 100,000 inhabitants.

**Figure 6 ijerph-17-08921-f006:**
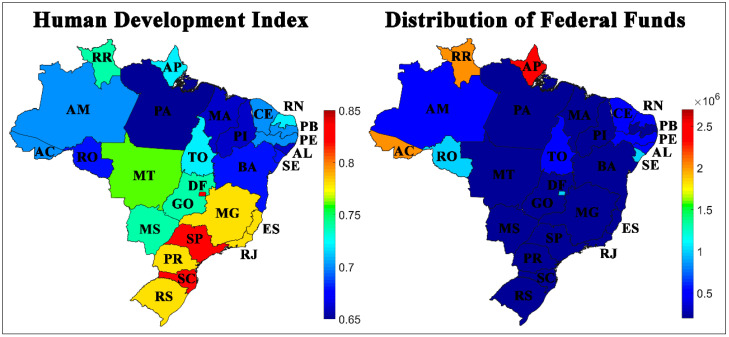
Geographical distribution by Index Development Index (HDI) and Federal funds destined to combat COVID-19 in Brazilian states per 100,000 inhabitants.

**Figure 7 ijerph-17-08921-f007:**
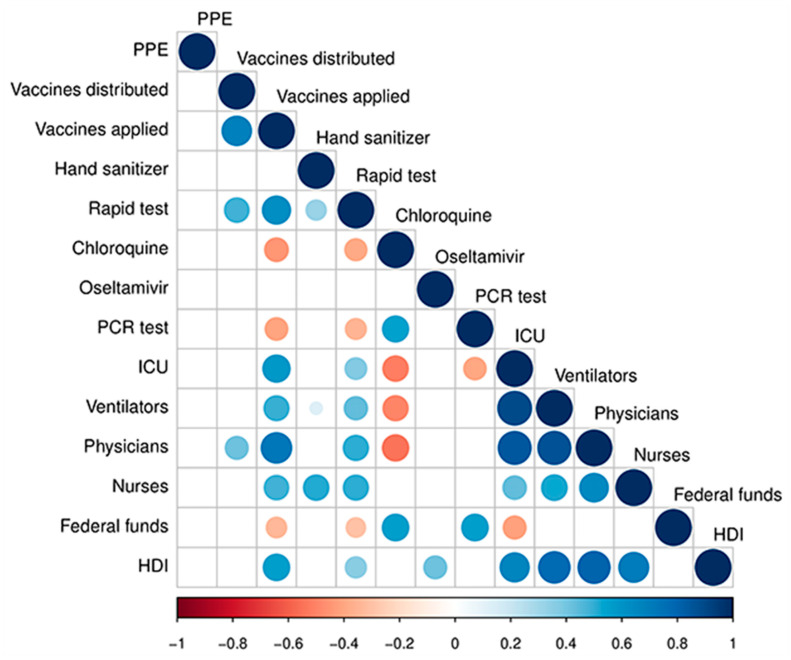
Spearman correlation coefficients for the 14 evaluated variables. Correlations with *p*-value > 0.05 are considered insignificant.

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
