# Peer review of "Can Socioeconomic, Health, and Safety Data Explain the Spread of COVID-19 Outbreak on Brazilian Federative Units?"

_ijerph, 2020, doi:10.3390/ijerph17238921_

Round 1
Reviewer 1 Report
This manuscript focused on the unsupervised analysis of 14 factors on the spread of COVID-19 on Brazilian, which meets the scope of International Journal of Environmental Research and Public Health. To improve the quality of the paper, some comments are as follows.
- Why the authors choose these 14 factors? Is there any literature to refer to? Will other significant factors be omitted?
- The authors cited very few references from this journal, which indicates the lack of adequate literature research.
- In Introduction, Line 51, the introduction of ANN network is too brief. The authors are encouraged to add more descriptions, such as supervised ANN, unsupervised ANN, and some up-to-date field applications, e.g. ANN-Based Continual Classification in Agriculture; Do we really need deep CNN for plant diseases identification; Few-shot cotton pest recognition and terminal realization.
- The Discussion Section should be treated seriously. In this version, the discussion seems to be missing.
- The experimental analysis needs more comparisons. The authors are suggested to compare another unsupervised algorithm, e.g. clustering, to testify the results.
Author Response
Dear reviewer, we appreciate your considerations about our work. Attached is the letter to the reviewers, while the changes made are highlighted throughout the manuscript.
Best regards,
D. Galvan and cols.

Reviewer 2 Report
My comments wish to improve how the author makes the paper attractive to the general audience and increase his impact. Right now is technically sound but not scientifically:
1).- Missing the fundamentals contribution of Artificial Neural Networks (ANN) (is recommended to justify theoretically, and it's recommended to add a section of theoretical considerations).
2).-Recommended to clarify why, is so important to study the Impact of Self-Organizing Maps (SOM) or Kohonen Map (classical methods based on mathematical theory versus black box methods based on modern ANN technology). One proposal is to add on the line 51:
Traditionally, exist several types of neural networks with various techniques such as Autoregression (AR) [1,2], Moving Average (MA) [3,4], Exponential Smoothing (ES) [5], Hybrid Methods (HM) [6,7,8] and Autoregressive Integrated Moving Average (ARIMA) [9] have been used to predict and forecast the dependent variable in a time series [1-9]. Among these techniques, unsupervised neural networks of the type Self-Organizing Maps model has mostly outperformed others in precision and accuracy [10].
[1] Jiang, C., Jiang, M., Xu, Q., Huang, X. (2017). Expectile regression neural network model with applications, Neurocomputing, 247, 73-86.
[2] Castañeda, A., Castaño, VM. (2017). Smart Frost Control in Greenhouses by Neural Networks Models. Computers and Electronics in Agriculture, 137, 102–114, ISSN: 0168-1699.
[3] Arora, S., Taylor, JW. (2018). Rule-based autoregressive moving average models for forecasting load on special days: A case study for France, European Journal of Operational Research, 266 (1), 259-268.
[4] Hassan, MM., Huda, S. Yearwood, J. Jelinek, HF., Almogren, A. (2018). Multistage fusion approaches based on a generative model and multivariate exponentially weighted moving average for diagnosis of cardiovascular autonomic nerve dysfunction, Information Fusion, 41, 105-118.
[5] Barrow, D., Kourentzes, N., Sandberg, R. Niklewski, J. (2020), Automatic robust estimation for exponential smoothing: Perspectives from statistics and machine learning, Expert Systems with Applications, 160, 113637.
[6] Baffour, AA., Feng, J., Taylor, EK. (2019), A hybrid artificial neural network-GJR modeling approach to forecasting currency exchange rate volatility, Neurocomputing, 365, 285-301.
[7] Castañeda, A., Castaño, VM. (2020). Smart frost measurement for anti-disaster intelligent control in greenhouses via embedding IoT and hybrid AI methods, Measurement, 164, 108043
[8] Pradeepkumar, D., Ravi, V. (2018). Soft computing hybrids for FOREX rate prediction: A comprehensive review, Computers and Operations Research, 99, 262-284.
[9] Castañeda-Miranda, A., Castaño, V.M. (2020). Internet of things for smart farming and frost intelligent control in greenhouses, Computers and Electronics in Agriculture, 176, 105614.
[10] Haykin, S. Neural Networks: A Comprehensive Foundation; Prentice Hal: New York, 2001; ISBN 978-0-02-352761-6.
Therefore, re-number the references and adapt the references format on the document. Also add more examples and use of SOM technique.
3).- In my own work in computer modelling, machine learning, etc., I have at times had to acknowledge that the results provided by complex technology systems are not necessarily statistically significantly better than the results that would be derived from a combination of descriptive statistics, simple sensors, and/or intuition. That is, can you be more explicit and quantify why & how (in what ways) this system is better (offers better data, more accurate data, more comprehensive data, etc.) than historical practices?. The possible answer is recommended to add it as a research hypothesis of the article.
4).- Please add equations that define the trend of behavior of the graphs and results presented.
5).- Its recommended to divide the paper into the sections: 1.-Introduction, 2.-Theoretical Foundations, 3.-Materials and Methods, 4.-Tests and Results, 5.-Conclusion & Future Work. The current structure of the article does not allow us to identify its importance, i.e., the manuscript is stronger at this document structure and should be revised before considering for publication.
6).- The literature survey is obsolete (all articles from 5 years ago are recommended) or is recommended update references (2015-2020 period), i.e., I think that several recent works in this field.
7).- Finally, grammar may be improved and equations, tables and pictures may be tuned and suited to the article.
Author Response

(The authors gave the same response as above.)

Reviewer 3 Report
The paper provides an interesting topic within the COVID19 infection in Brasil, while it turned out to be a pandemic.
Overall, I think the paper is well written and well analyzed. Apart from minor English editing, I suggest the following actions before a final version should be submitted:
- Go carefully over the insights and the consequences for policy recommendations. You now mention things that could be interpreted in many different ways. Is it good to have these measures? That you probably cannot tell from your research. But you can tell what to look for and the parameters to observe when putting in restrictions.
- The reliability of the data set should have verification, especially when you are talking about Health and Safety, any attempt or assessment made before?
- Go over the generalisability of your work: You did a post-assessment. What if this will hold true for other viruses or the same virus in other regions?
Author Response

(The authors gave the same response as above.)

Reviewer 4 Report
This is a very interesting and potentially important report. However, the mixture of description (Result) and analysis/interpretation/contextualisation (implications of the Result) in the Results section and the insertion of what appears to be a reviewers comment as the only content in the Discussion section renders the manuscript unsuitable for publication in its present form.
The authors should reorganise the Results section to exclude analysis/interpretation/contextualisation and include only description of the Result. The analysis/interpretation/contextualisation of each Result should be presented in a full Discussion section and interpreted according to what appears to be the advice in the earlier reviewers comment presented as 4. Discussion in the current manuscript i.e.:
"Authors should discuss the results and how they can be interpreted in perspective of previous studies and of the working hypotheses. The findings and their implications should be discussed in the broadest context possible. Future research directions may also be highlighted."
Some small matters;
Line 32: The novel coronavirus descriptor is SARS-CoV-2. COVID 19 is the disease condition caused by this virus. Suggest add "SARS-CoV-2, the causative agent of" after the word 'against'
Line 219: Antiviral drugs target the infectious agent, so SARS-CoV-2 is the better descriptor in this line and in other places in the manuscript directly referencing the causative agent.
Author Response

(The authors gave the same response as above.)

Round 2
Reviewer 1 Report
- The Discussion section is not sufficient. The authors are suggested to discuss the impact of parameters of output layer of SOM on the conducted results, e.g., size of output layer, shape of output unit, similarity measures.
- The added comparative experiments in the responsed letter with clustering algorithm cannot be found in this revised version, please add them.
- The references are too many, please consider simplifying.
Author Response
Dear reviewer,
We appreciate your contributions. Attached is a letter with the highlighted information that was inserted in the manuscript, discussing point by point each of the suggested notes. We hope we have been able to answer your questions.
Best regards,
D. Galvan and cols.

Reviewer 4 Report
Thank you for the response to comments. You have satisfactorily addressed the original concerns raised.
Author Response
Dear Reviewer,
We appreciate your contributions. As suggested, minor improvements and spelling corrections were made throughout the text.
Best regards,
D. Galvan and cols.